# Time Trends in Peer Violence and Bullying Across Countries and Regions of Europe, Central Asia, and Canada Among Students Aged 11, 13, and 15 from 2013 to 2022

**DOI:** 10.3390/bs16010036

**Published:** 2025-12-24

**Authors:** Gabriele Prati

**Affiliations:** Department of Psychology, University of Bologna, Piazza Aldo Moro, 90, 47521 Cesena (FC), Italy; gabriele.prati@unibo.it

**Keywords:** aggression, school-age child, gender, bullying, cyberbullying, COVID-19 pandemic

## Abstract

Background: The impact of the COVID-19 pandemic on temporal trends in peer violence and bullying deserves closer scrutiny. The aim of the present study was to examine temporal trends in peer violence and bullying among school-aged children before and after the COVID-19 pandemic. Methods: Data from the Health Behaviour in School-aged Children (HBSC) surveys (2013/2014–2021/2022) were analyzed to track changes in peer violence and bullying over time. The sample encompassed over 700,000 students aged 11, 13, and 15 from more than 40 countries across Asia, Europe, and North America. Results: Traditional (school) bullying perpetration and victimization did not change significantly over time. A significant decreasing trend in engagement in physical fighting between the 2013/2014 and 2021/2022 surveys was observed among male participants aged 15. In contrast, a significant increasing trend in engagement in physical fighting was observed among female participants aged 11 and 13 years. Following the pandemic, increases in cyberbullying perpetration and victimization were observed among students aged 11 and 13, a trend not evident among 15-year-olds. Conclusion: Except for cyberbullying, the pandemic did not appear to influence trends in peer violence and bullying, which remained largely stable or reflected trajectories that had begun prior to the pandemic.

## 1. Introduction

The outbreak of the COVID-19 pandemic was associated with unprecedented health, economic, and social disruptions worldwide. While the impact of the COVID-19 pandemic on adult mental health was small in magnitude ([26]), young people’s mental health and well-being appear to have been more adversely affected ([7]; [20]; [22]). To designate mental health symptoms occurring in childhood and adolescence, the terms internalizing and externalizing problems have traditionally been adopted ([19]). Externalizing behaviors are characterized primarily by outward-directed actions, such as acting out, aggression, and antisocial or criminal behavior. Although a great deal of research has focused on youth mental health and well-being, less is known about changes in aggressive and bullying behavior, as well as victimization, following the COVID-19 pandemic ([35]).

The prevention and protection measures implemented to contain the COVID-19 pandemic reduced peer contact. Given that aggressive and bullying behavior typically involves peers, the COVID-19 pandemic might have decreased opportunities to engage in such behaviors. A review of studies investigating changes in bullying prevalence rates across the world during the COVID-19 pandemic revealed that many countries experienced reductions in bullying victimization and perpetration ([35]). However, in countries with fewer social restrictions, increases in rates of bullying were also observed. This increase in bullying victimization was not supported by a meta-analysis of 79 studies, which revealed a significant decrease in rates of traditional bullying victimization during the COVID-19 pandemic years (2020–2022) compared with the pre-pandemic years ([18]). Additionally, a meta-analysis on cyberbullying revealed that the prevalence of overall cyberbullying, victimization, and perpetration during the COVID-19 pandemic was lower than before the COVID-19 pandemic ([12]). In contrast, another meta-analysis found that rates of cyberbullying victimization during the COVID pandemic years of 2020 to 2022 were similar to preceding years ([18]). To complicate matters further, a systematic review highlighted opposite trends: a decrease in the prevalence of cyberbullying and/or cybervictimization in Western countries, but an increase in many Asian countries and in Australia between 2020 and 2023 ([31]). Taken together, these reviews and meta-analyses present mixed findings concerning changes in bullying prevalence rates during the COVID-19 pandemic.

A critical distinction lies between changes that occurred during the COVID-19 pandemic and those that emerged after the pandemic, following the return to school and other routine activities. During the COVID-19 pandemic, the associated containment measures (e.g., lockdowns) had a substantial impact on daily routines. For instance, school closures and quarantine-related restrictions prohibited or greatly limited in-person schooling, after-school programs, and leisure activities, which, in turn, impacted socialization processes and peer group interactions. Based on routine activities theory ([4]), such altered daily routines were expected to transform the dynamics of bullying and peer violence. Furthermore, the COVID-19 pandemic and the related containment measures have been conceptualized as a form of repeated psychosocial strain that, in turn, affects deviant behavior among young people ([8]; [28]), in line with the propositions of the general strain theory ([1], [2]). However, it remains unclear whether the strain caused by the COVID-19 pandemic should be expected to have short-term effects (i.e., during the pandemic) or long-term effects (i.e., after the pandemic). [25] ([25]) found that the rates of most crime types remained largely stable throughout and following the COVID-19 pandemic. A decrease in child sexual abuse, theft, and drug trafficking and possession was observed only during the pandemic period. After the pandemic, an increase in attempted murder, physical assault, deliberate injury, threats, and robbery was observed, whereas other types of crime remained unaltered. Another study conducted after the COVID-19 pandemic found that the relationship between days of school closure and school bullying and cyberbullying perpetration and victimization was negative, whereas the relationship with physical fighting was non-significant ([36]). The findings of this study suggest that national policies surrounding school closures during the COVID-19 pandemic might not constitute a repeated psychosocial strain capable of increasing peer violence among young people. However, this study did not investigate a potential change in peer violence before and after the COVID-19 pandemic. Moreover, [36] ([36]) found that the effects were primarily observed among boys rather than girls, suggesting the opportunity to apply a gendered lens to understand the potential effects of the COVID-19 pandemic ([11]). Finally, [36] ([36]) documented the strongest effects of school closures among older adolescents, thereby indicating the need to consider age groups separately.

### Purpose of the Present Study

The main aim of the present study was to examine time trends in peer violence and bullying among school-aged children before and after the COVID-19 pandemic. Physical fighting, as well as traditional bullying perpetration and victimization, were used as indicators of peer violence and bullying. Furthermore, to enhance generalizability, data from a wide range of countries were included. Finally, the data were analyzed separately by gender and age group to investigate potential changes from a gendered perspective.

## 2. Method

### 2.1. Participants and Procedure

This study draws on data from the Health Behaviour in School-aged Children (HBSC) survey, a large, quadrennial, cross-sectional, school-based initiative across Europe, Asia, and North America. Data were collected using a standardized protocol ([14]) from students aged 11, 13, and 15 years. The 2013/2014 HBSC survey comprises data from 219,460 young people in 42 countries and regions. The 2017/2018 HBSC contains information for 227,441 young people from 45 countries and regions. The 2021/2022 HBSC includes data for 279,117 young people from 44 countries and regions. To achieve a sample representative of the full age range, each participating country or region employed cluster sampling to select a proportionate number of participants aged 11, 13, and 15 years. Cluster sampling was used to draw samples, with entire schools or school classes serving as the primary sampling units. The HBSC study secured ethical approval from the Institutional Review Board (IRB) or equivalent ethics committee of the corresponding university or research center in every participating country. A detailed description of the ethical procedures, conceptual framework, and survey methodology is available on the HBSC website (https://hbsc.org/) and related publications (e.g., [14]). All measures were collected in the three survey waves, except for data on cyberbullying perpetration and victimization, which were gathered only in the 2017/2018 and 2021/2022 HBSC surveys. All available data were used, and no country or respondent was excluded due to incomplete information.

### 2.2. Instrument

#### 2.2.1. Bullying Perpetration and Victimization

In the HBSC, two items from the Olweus Bullying Questionnaire ([21]) were adapted to measure traditional bullying perpetration and victimization. Specifically, participants were asked how often they had taken part in bullying (an) other person(s) [or they had been bullied by (an) other person(s)] at school in the past couple of months. At the beginning of the item set, an explanation of what constitutes bullying was provided to operationalize the concept for participants. Response options ranged from never to several times a week. In line with the HBSC methodology (e.g., [14]) and previous research (e.g., [36]), a cut-off of “2–3 times a month or more” was used to identify a regular pattern of engagement or victimization.

#### 2.2.2. Cyberbullying Perpetration and Victimization

Participants were asked whether they had taken part in cyberbullying (e.g., sent mean instant messages, emails, or text messages; posted hurtful comments; created a website making fun of someone; posted or shared photos or videos online without permission) in the past couple of months to measure perpetration. To measure victimization, respondents were asked to indicate how often they had been cyberbullied (with the same examples). Response options ranged from never to several times a week. Following the HBSC methodology (e.g., [14]) and in line with previous research (e.g., [5]), a cut-off of “at least once or twice in the past couple of months” was used.

#### 2.2.3. Physical Fighting

Physical fighting was included as it represents a form of peer violence characterized by a relatively equal balance of power between those who engage in this behavior. The study asked participants to state the number of times they had been involved in a physical fight during the past year. Response options ranged from none to four times or more. Following the guidance from the HBSC methodology (e.g., [14]) and established research (e.g., [36]), a cut-off of three times or more was used to identify recurrent engagement in physical fighting.

### 2.3. Data Analysis

Data analyses were conducted using R software (version 4.5.0). Proportions for each behavior across countries, obtained from the [9] ([9]), were used in the analyses. The Kruskal–Wallis test was applied to examine differences among the 2013/2014, 2017/2018, and 2021/2022 HBSC surveys. For the Kruskal–Wallis test, the eta-squared measure (η^2^) based on the H statistic was computed using the formula described by [33] ([33]). The eta-squared statistic represents the proportion of variance in the dependent variable explained by the independent variable. According to [13] ([13]), an effect size of approximately 0.01 can be considered small, around 0.09 medium, and 0.25 or larger large. Dunn’s test was employed as a post hoc procedure following the Kruskal–Wallis test. Holm’s method was applied as a multiple comparison adjustment to Dunn’s test. The Kruskal–Wallis test, its associated effect size, and Dunn’s post hoc comparisons were computed using the *R* package *rstatix* ([15]). To determine the presence of a statistically significant trend in the data (i.e., ordered pattern), the Jonckheere–Terpstra test was conducted using the *R* package *DescTools* ([30]).

## 3. Results

The median and interquartile range for study variables across age, year, and gender are presented in Table 1 and Figure 1.

### 3.1. Physical Fighting

Among female students, the proportion of physical fighting differed significantly across the three surveys for participants aged 11, H(2) = 20.2, *p* < .001, η^2^ = 0.15, or aged 13 years, H(2) = 10.8, *p* = .005, η^2^ = 0.07. However, for female participants aged 15 years, the proportion of physical fighting did not differ significantly across the three surveys, H(2) = 1.2, *p* = .562, η^2^ = 0.01. Post hoc comparisons using Dunn’s test indicated no statistically significant differences between the 2013/2014 and 2017/2018 surveys among female participants aged 11 years, *z* = 1.63, *p* = .104, and 13, *z* = 1.52, *p* = .158. Moreover, statistically significant differences were found between the 2013/2014 and 2021/2022 surveys among female participants aged 11, *z* = 4.43, *p* < .001, or aged 13 years, *z* = 3.27, *p* = .003. Finally, the difference between the 2017/2018 and 2021/2022 surveys was statistically significant among female participants aged 11 years, *z* = 2.81, *p* = .009, but not among those aged 13 years, *z* = 1.76, *p* = .158. The Jonckheere–Terpstra test revealed a statistically significant and increasing trend among female participants aged 11 years, JT = 3678, *p* = .002, and 13, JT = 3409, *p* = .002.

Among male participants, the proportion of physical fighting was not significantly different across the three surveys for participants aged 11, H(2) = 2.3, *p* = .324, η^2^ = 0.00, or aged 13 years, H(2) = 2.3, *p* = .312, η^2^ = 0.00. Conversely, the proportion of physical fighting differed significantly across the three surveys, H(2) = 6.3, *p* = .042, η^2^ = 0.03, among participants aged 15 years. Post hoc comparisons revealed no statistically significant differences between the 2013/2014 and 2017/2018 surveys, *z* = −0.93, *p* = .353, and between the 2017/2018 and 2021/2022 surveys, *z* = −1.56, *p* = .236. There was a statistically significant difference between the 2013/2014 and the 2021/2022 surveys, *z* = −2.48, *p* = .013, among male participants aged 15 years. The Jonckheere–Terpstra statistic tests indicated a statistically significant and decreasing trend among male participants aged 15 years, JT = 2071, *p* = .014.

### 3.2. Bullying Perpetration

Among female participants, the proportion of bullying perpetration did not change across the three surveys for participants aged 11, H(2) = 1.9, *p* = .371, η^2^ = 0.00, aged 13 years, H(2) = 2.0, *p* = .366, η^2^ = 0.00, or aged 15 years, H(2) = 2.9, *p* = .232, η^2^ = 0.01. Among male students, the proportion of bullying perpetration was not significantly different across the three surveys for participants aged 11, H(2) = 4.5, *p* = .106, η^2^ = 0.02, aged 13, H(2) = 4.6, *p* = 0.102, η^2^ = 0.02, or aged 15 years, H(2) = 3.1, *p* = .208, η^2^ = 0.01.

### 3.3. Bullying Victimization

The proportion of bullying victimization did not change across the three surveys for female participants aged 11, H(2) = 3.7, *p* = .160, η^2^ = 0.01, aged 13 years, H(2) = 3.6, *p* = .162, η^2^ = 0.01, or aged 15 years, H(2) = 1.2, *p* = .558, η^2^ = 0.00. Similarly, among male students, no differences were found across the three surveys for participants aged 11, H(2) = 2.0, *p* = .365, η^2^ = 0.00, aged 13 years, H(2) = 1.4, *p* = .487, η^2^ = 0.00, or 15 years, H(2) = 2.1, *p* = .357, η^2^ = 0.00.

### 3.4. Cyberbullying Perpetration

Among female students, there was a statistically significant increase in the proportion of cyberbullying perpetration between the two surveys for participants aged 11, H(2) = 10.5, *p* = .001, η^2^ = 0.11, or aged 13 years, H(2) = 5.9, *p* = .015, η^2^ = 0.06. However, for female participants aged 15 years, the proportion of cyberbullying perpetration was not significantly different across the two surveys, H(2) = 0.5, *p* = .495, η^2^ = 0.00. A similar pattern was observed among male participants. Specifically, the increase in the proportion of cyberbullying perpetration between the two surveys was statistically significant for participants aged 11, H(2) = 7.8, *p* = .005, η^2^ = 0.08, or aged 13 years, H(2) = 8.6, *p* = .003, η^2^ = 0.09, but not for those aged 15 years, H(2) = 1.8, *p* = .178, η^2^ = 0.01.

### 3.5. Cyberbullying Victimization

Among female participants, the tests revealed statistically significant increases in the proportion of cyberbullying victimization between the two surveys for students aged 11, H(2) = 13.4, *p* < .001, η^2^ = 0.15, or aged 13 years, H(2) = 5.9, *p* = .015, η^2^ = 0.06. However, for female participants aged 15 years, the proportion of cyberbullying victimization did not change significantly between the two surveys, H(2) = 1.0, *p* = .323, η^2^ = 0.00. Again, similar results were obtained for male participants. Specifically, the increase in the proportion of cyberbullying perpetration between the two surveys was statistically significant for participants aged 11, H(2) = 9.1, *p* = .002, η^2^ = 0.10, or aged 13 years, H(2) = 8.3, *p* = .004, η^2^ = 0.09, but not for those aged 15 years, H(2) = 3.2, *p* = .071, η^2^ = 0.02.

## 4. Discussion

The aim of this manuscript was to examine temporal trends in peer violence and bullying among school-aged children before and after the COVID-19 pandemic. Overall, the results indicated that traditional (school) bullying perpetration and victimization did not differ significantly between the pre-pandemic and post-pandemic periods. Moreover, engagement in physical fighting between 2013/2014 and the 2021/2022 surveys increased among female participants aged 11 and 13 years and decreased among male participants aged 15 years. The results suggested that these changes could partly be attributed to trends that began before the COVID-19 pandemic. Finally, cyberbullying perpetration and victimization increased following the pandemic among students aged 11 and 13 years, but not among those aged 15 years. Given the reliance on only two time points (i.e., before and after the pandemic), it was not possible to test whether trends had begun prior to the COVID-19 pandemic.

[17] ([17]) found increasing trends in cyberbullying and decreasing trends in traditional bullying perpetration and victimization from 1998 to 2017. The findings of this study suggest that the increased awareness of traditional bullying and the implementation of bullying prevention programs, which generated such a decline, have reached a plateau. The continued increase in cyberbullying highlights the need for enhanced strategies targeting online bullying prevention.

Previous research has highlighted a decrease in the prevalence of bullying victimization during or immediately after the COVID-19 lockdowns (e.g., [12]; [18]; [23]; [29]; [34]; [35]). The COVID-19 pandemic might have had a short-term reducing effect on bullying and peer victimization, partly due to containment measures. The rates of traditional bullying and cyberbullying victimization dropped in 2020, with a gradual return to pre-COVID rates in 2021 and 2022 as students returned to school ([18]; [23]). As data in the current study were collected in 2021/2022 following the return to in-person schooling, the findings cannot rule out such a short-term effect. The current study demonstrated that the prevalence of traditional (school) bullying perpetration and victimization after the COVID-19 pandemic was consistent with the pre-COVID period, whereas an increase in cyberbullying perpetration and victimization was observed among students aged 11 and 13 years. This trend may stem from a combination of diminished oversight from parents and schools, the effects of online disinhibition, and specific developmental risks concerning students aged 11 and 13 years. For instance, younger students may invest more emotional significance in their digital social circles, a factor that can increase both their likelihood of encountering cyber-aggression and their psychological vulnerability to it. Moreover, unlike traditional bullying, the digital nature of cyberbullying allows a single incident to persist and proliferate through repeated sharing. This enduring visibility likely explains why online victimization rebounded so rapidly following the easing of social restrictions; the digital infrastructure for aggression remained active and scalable even when physical contact was limited ([16]). Future studies might investigate potential long-term effects of the COVID-19 pandemic. Compared with the COVID and pre-COVID periods, evidence suggests an increase in violent crimes (e.g., attempted murder, deliberate injury, physical assault) among minors in 2023 ([25]).

The findings of this study suggest an increase in the early onset of peer violence among female students. Previous research highlighted that early onset of aggressive behavior is a risk factor that increases the likelihood of violence during adolescence and young adulthood (e.g., [6]). In addition, an early onset of aggressive behavior can exacerbate the association between pubertal development and later aggressive behavior ([27]).

The present study did not examine cross-country contextual moderators (e.g., variation in school closures, cultural norms, or digital access). A previous cross-national study showed that national-level lockdown measures—specifically, the number of days of school closure—were negatively associated with several forms of bullying perpetration and victimization, with the exception of physical fighting. However, the magnitude of this association was small according to the criteria proposed by [3] ([3]). Moreover, the effect was observed only when comparing countries in the highest tercile of school closures (141–230 days) with those in the lowest tercile (fewer than 90 days), whereas comparisons between the medium and low tercile groups were nonsignificant. These findings suggest that the role of school closures as a cross-country contextual moderator is limited. Although cultural norms and digital access may also function as cross-country contextual moderators, a clear theoretical framework specifying their mechanisms and expected effects on the outcomes examined in the present study has yet to be developed.

### 4.1. Implications for Practice

An important first step toward preventing bullying and peer violence is the identification of the factors that affect the risk of violent victimization and perpetration (e.g., [6]). A history of early aggression is an important individual factor. Although male students are generally more likely than female students to engage in physical fighting, the findings of this study suggest a need to focus on female students aged 11 and 13 years, who exhibit an increase in peer violence. In addition, the prevalence of face-to-face forms of bullying has remained stable, whereas an increase in cyberbullying perpetration and victimization was observed among students aged 11 and 13 years. The increase in Internet use may have been accompanied by the emergence of new spaces for peer violence. School-based programs appear to be effective in preventing and addressing cyberbullying ([32]). Finally, the prevention of youth violence might take into account the role played by ecological factors ([10]). For instance, promoting a sense of school community can be regarded as a means of reducing students’ aggressive behavior, particularly in schools with a higher prevalence of violence ([24]). Furthermore, individual, peer, and family intervention approaches have been used to reduce aggressive behavior among students (e.g., [6]).

### 4.2. Limitations and Future Research Directions

The observational nature of the study makes it difficult to draw causal inferences regarding the impact of the COVID-19 pandemic. Although the effects of third variables cannot be ruled out, the COVID-19 pandemic was the most significant event during 2020–2021, and it is unlikely that any other event had a greater impact. In addition, the third survey was conducted in 2021/2022. Therefore, the present study cannot provide information about the immediate or long-term (i.e., post-2022) effects of the COVID-19 pandemic. Future studies might examine the long-term effects of the COVID-19 pandemic. Furthermore, the self-reported nature of the data collection may have introduced response biases. However, it seems unlikely that these biases differentially affected the findings across the surveys. The current study reported data from European, Central Asian, and North American regions. Therefore, caution is warranted when interpreting or generalizing the results. Future research including countries from other regions of the world (e.g., Central and South America, Africa) is warranted. In this study, each subgroup was analyzed separately. It is well known that conducting multiple statistical tests can inflate the Type I error rate. To address this issue, the Holm adjustment was applied to control the Type I error rate for multiple testing. Finally, compared with cyberbullying, a stricter criterion was used to identify patterns of traditional (face-to-face) bullying, following the standardized protocol of the HBSC study and the Olweus definition, which requires a certain degree of repetition and frequency to distinguish bullying from isolated incidents. This difference in thresholds is justified by the HBSC methodology and by previous literature on cyberbullying (e.g., [5]). In the context of cyberbullying, a single act is often considered to have repetitive and lasting effects due to the nature of online communication. As a result, this discrepancy in inclusion criteria limits the direct comparability of prevalence rates between the two forms of bullying, as victimization is defined more broadly for cyberbullying than for traditional bullying. However, direct comparisons of prevalence rates between cyberbullying and traditional bullying were not the primary aim of the present study. Instead, the focus was on examining changes in prevalence over time within the same form of perpetration or victimization.

## 5. Conclusions

The prevalence of traditional (school) bullying perpetration and victimization did not appear to be affected by the COVID-19 pandemic and has continued to represent a public health concern. In addition, the prevalence of cyberbullying perpetration and victimization increased following the pandemic among students aged 11 and 13 years. This finding may be explained by greater exposure to online environments at earlier ages, particularly as a result of containment measures related to the COVID-19 pandemic. Finally, an increasing trend in physical fighting between the 2013/2014 and 2021/2022 surveys was observed among female participants aged 11 and 13 years, whereas the prevalence of this behavior has significantly declined over time among male participants aged 15 years. These trends warrant continued monitoring in future research.

## Figures and Tables

**Figure 1 behavsci-16-00036-f001:**
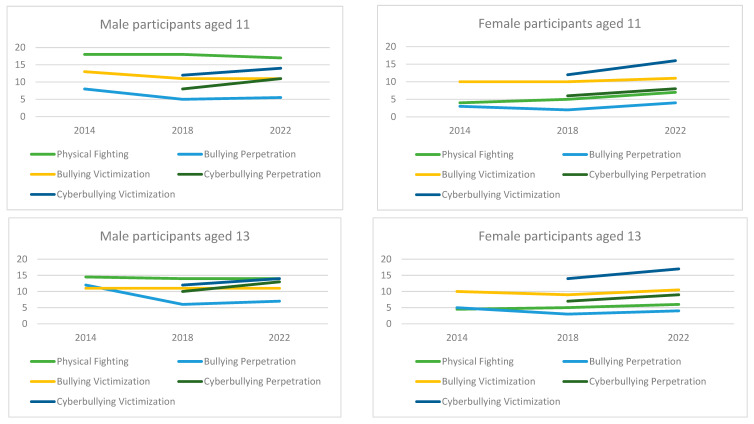
Median for the Proportion of Peer Violence and Peer Bullying Across Age, Year, and Gender.

**Table 1 behavsci-16-00036-t001:** Median (Mdn) and Interquartile Range (IQR) for the Proportion of Peer Violence and Peer Bullying Across Age, Year, and Gender.

Age	Year	Gender	Physical Fighting		Bullying Perpetration		Bullying Victimization		Cyberbullying Perpetration		Cyberbullying Victimization	
			*Mdn*	*IQR*	*Mdn*	*IQR*	*Mdn*	*IQR*	*Mdn*	*IQR*	*Mdn*	*IQR*
11	2014	Male	18.0	5.3	8.0	9.5	13.0	7.5	—	—	—	—
11	2018	Male	18.0	5.8	5.0	9.0	11.0	6.0	8.0	9.0	12.0	7.0
11	2022	Male	17.0	6.0	5.5	7.3	11.0	6.3	11.0	8.3	14.0	6.3
13	2014	Male	14.5	6.5	12.0	9.0	11.0	7.0	—	—	—	—
13	2018	Male	14.0	4.8	6.0	7.0	11.0	7.0	10.0	7.0	12.0	8.0
13	2022	Male	14.0	5.0	7.0	6.3	11.0	6.0	13.0	6.3	14.0	5.5
15	2014	Male	13.0	4.0	10.0	9.5	9.0	5.5	—	—	—	—
15	2018	Male	12.0	3.8	8.0	5.0	7.0	5.0	12.0	8.0	11.0	7.0
15	2022	Male	10.0	4.0	8.5	5.0	8.0	6.0	15.0	5.8	13.0	5.5
11	2014	Female	4.0	1.3	3.0	5.0	10.0	6.0	—	—	—	—
11	2018	Female	5.0	3.8	2.0	4.0	10.0	5.0	6.0	4.0	12.0	7.0
11	2022	Female	7.0	3.0	4.0	5.0	11.0	5.0	8.0	5.0	16.0	7.0
13	2014	Female	4.5	3.0	5.0	7.5	10.0	6.0	—	—	—	—
13	2018	Female	5.0	3.0	3.0	4.0	9.0	6.0	7.0	5.0	14.0	9.0
13	2022	Female	6.0	2.0	4.0	4.5	10.5	6.3	9.0	5.0	17.0	8.3
15	2014	Female	4.0	2.0	4.0	5.0	8.0	4.0	—	—	—	—
15	2018	Female	4.0	2.8	3.0	3.0	6.0	3.0	7.0	5.0	12.0	7.0
15	2022	Female	4.0	3.0	3.0	4.0	7.0	5.0	7.5	3.3	13.5	5.5

*Note.* Data on cyberbullying perpetration and victimization were collected only in the 2017/2018 and 2021/2022 HBSC surveys.

## Data Availability

Data are available on the HBSC website (https://hbsc.org/).

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
