# Peer review of "Time Trends in Peer Violence and Bullying Across Countries and Regions of Europe, Central Asia, and Canada Among Students Aged 11, 13, and 15 from 2013 to 2022"

_behavsci, 2025, doi:10.3390/bs16010036_

Round 1

Reviewer 1 Report

Comments and Suggestions for Authors

In this paper, the author examined how bullying, physical fighting, and cyberbullying among adolescents have evolved before and after the COVID-19 pandemic. The paper is impressive and I really enjoyed reading this paper a lot. The topic is timely and important. Using HBSC data from over 700,000 students across 40 countries is a real strength, and the global scope is impressive. I agree it will contribute well to the literature. Here are some of my comments to improve the manuscript further:

1. First, there might be some conceptual blurring between “during” and “after” COVID-19. Since the dataset spans 2013/14, 2017/18, and 2021/22, none of these waves captures the lockdown period itself. The text should clarify that this study captures post-pandemic trends rather than direct effects during lockdowns.

2. The introduction would benefit from more discussion of cross-country contextual moderators (e.g., varying school closures, cultural norms, digital access) given the cross-national scope

3. I might be wrong but in the description of physical fighting, the item is said to ask about the number of fights during the past year, yet the cut-off is described as ‘2–3 times a month or more’. This might be inconsistent with the stated response format and with standard HBSC coding, and it is unclear whether this is a wording error or reflects how the variable was actually coded. It will be nice for the authors to just check again. 

4. Missing data are not described. I would like to encourage the author to report how many countries or respondents were excluded due to incomplete information?

5. Figures showing time trends (e.g., line plots by age and gender) would make the results much easier to follow.

6. The finding on the cyberbullying trend is particularly important, and I would like to encourage the author to expand the discussion further. While the manuscript notes that cyberbullying increased among younger girls after the pandemic, the explanation could be developed in greater depth. This pattern may not only reflect increased online activity but also reduced parental and school supervision, heightened online disinhibition, and developmental or gender-related vulnerabilities. Younger girls often place stronger emotional value on their online relationships, which can heighten both exposure to and sensitivity toward digital aggression. Also, I think unlike traditional bullying, cyberbullying can persist and spread over time as a single post or message is repeatedly shared, which may help explain why online victimization rebounded more quickly once social restrictions eased. A more in-depth discussion of these mechanisms would help clarify this important finding. See the following review that can help to build the argument: Kasturiratna et al. (2025). Umbrella review of meta-analyses on the risk factors, protective factors, consequences and interventions of cyberbullying victimization. Nature Human Behaviour, 9(1), 101-132.

Author Response

Reviewer 1

In this paper, the author examined how bullying, physical fighting, and cyberbullying among adolescents have evolved before and after the COVID-19 pandemic. The paper is impressive and I really enjoyed reading this paper a lot. The topic is timely and important. Using HBSC data from over 700,000 students across 40 countries is a real strength, and the global scope is impressive. I agree it will contribute well to the literature. Here are some of my comments to improve the manuscript further:

  1. First, there might be some conceptual blurring between “during” and “after” COVID-19. Since the dataset spans 2013/14, 2017/18, and 2021/22, none of these waves captures the lockdown period itself. The text should clarify that this study captures post-pandemic trends rather than direct effects during lockdowns.

Response: We agree that it is important to clarify that this study did not capture peer violence and bullying during the lockdown period. We have revised the manuscript to make clear that this study examined temporal trends in peer violence and bullying among school-aged children before and after the COVID-19 pandemic.

  1. The introduction would benefit from more discussion of cross-country contextual moderators (e.g., varying school closures, cultural norms, digital access) given the cross-national scope

Response: Following your suggestion, we added a discussion of cross-country contextual moderators. While there are some theory and evidence concerning varying school closures, a clear theoretical framework specifying mechanisms of cultural norms and digital access and their expected moderation effects on the outcomes examined in the present study has yet to be developed.

  1. I might be wrong but in the description of physical fighting, the item is said to ask about the number of fights during the past year, yet the cut-off is described as ‘2–3 times a month or more’. This might be inconsistent with the stated response format and with standard HBSC coding, and it is unclear whether this is a wording error or reflects how the variable was actually coded. It will be nice for the authors to just check again.

Response: Thank you for catching this error. The response format is now consistent with standard HBSC coding.

  1. Missing data are not described. I would like to encourage the author to report how many countries or respondents were excluded due to incomplete information?

Response: Thank you for asking this question. We have clarified that no country or respondent was excluded due to incomplete information.

  1. Figures showing time trends (e.g., line plots by age and gender) would make the results much easier to follow.

Response: Following your request, a figure showing time trends was added.

  1. The finding on the cyberbullying trend is particularly important, and I would like to encourage the author to expand the discussion further. While the manuscript notes that cyberbullying increased among younger girls after the pandemic, the explanation could be developed in greater depth. This pattern may not only reflect increased online activity but also reduced parental and school supervision, heightened online disinhibition, and developmental or gender-related vulnerabilities. Younger girls often place stronger emotional value on their online relationships, which can heighten both exposure to and sensitivity toward digital aggression. Also, I think unlike traditional bullying, cyberbullying can persist and spread over time as a single post or message is repeatedly shared, which may help explain why online victimization rebounded more quickly once social restrictions eased. A more in-depth discussion of these mechanisms would help clarify this important finding. See the following review that can help to build the argument: Kasturiratna et al. (2025). Umbrella review of meta-analyses on the risk factors, protective factors, consequences and interventions of cyberbullying victimization. Nature Human Behaviour, 9(1), 101-132.

Response: Thank you for your comments. We would like to clarify that the findings related to cyberbullying are more nuanced and complex and cyberbullying has increased among younger girls as well as among younger boys. Specifically, among female students, there was a statistically significant increase in the proportion of cyberbullying perpetration between the two surveys for participants aged 11 and 13 years. In contrast, for female participants aged 15 years, the proportion of cyberbullying perpetration did not differ significantly across the two surveys. A similar pattern was observed among male participants: statistically significant increases in cyberbullying perpetration were found for those aged 11 and 13 years, but not for those aged 15 years. With regard to cyberbullying victimization, statistically significant increases between the two surveys were observed among female participants aged 11 and 13 years, whereas no significant change was detected among female participants aged 15 years. Again, comparable results were obtained for male participants, with significant increases in victimization observed among younger adolescents (ages 11 and 13) but not among those aged 15 years.

Following your request, we have expanded the Discussion section to better reflect these age-specific patterns and to clarify the complexity of the findings.

Reviewer 2 Report

Comments and Suggestions for Authors

For Traditional School Bullying:
On lines 128-129, the author establishes a cutoff point of "2-3 times a month or more."
Context: This stricter criterion is used to identify patterns of "traditional bullying" (face-to-face), following the standardized protocol of the HBSC study and the Olweus definition, which requires a certain degree of repetition and frequency to distinguish it from isolated incidents.
For Cyberbullying:
On lines 137-138, the criterion is much more lenient: "at least once or twice in the past couple of months."
Context: This lower threshold is also justified based on the HBSC methodology and previous literature on cyberbullying. In cyberbullying, a single act (such as posting a humiliating photo) is often considered to have repetitive and lasting effects due to the nature of the internet. Therefore, the same frequency of occurrence as in physical bullying is not always required to be considered significant.
It is crucial that the author clarify in the Discussion whether this difference in inclusion criteria affects the direct comparison of prevalence rates for both types of violence, since "being a victim" is defined much more broadly in cyberbullying than in traditional bullying in this study.

The author uses the Kruskal-Wallis test to compare the three waves (2014, 2018, 2022) and the Jonckheere-Terpstra test for ordered trends.
Although these tests are robust for nonnormal distributions, they are limited in detecting complex interactions (e.g., Year ≥ Gender ≥ Age) that are central to the study hypothesis. 
The manuscript analyzes each subgroup separately, inflating the Type I error (although it uses Holm correction). 

Author Response

For Traditional School Bullying:

On lines 128-129, the author establishes a cutoff point of "2-3 times a month or more."

Context: This stricter criterion is used to identify patterns of "traditional bullying" (face-to-face), following the standardized protocol of the HBSC study and the Olweus definition, which requires a certain degree of repetition and frequency to distinguish it from isolated incidents.

For Cyberbullying:

On lines 137-138, the criterion is much more lenient: "at least once or twice in the past couple of months."

Context: This lower threshold is also justified based on the HBSC methodology and previous literature on cyberbullying. In cyberbullying, a single act (such as posting a humiliating photo) is often considered to have repetitive and lasting effects due to the nature of the internet. Therefore, the same frequency of occurrence as in physical bullying is not always required to be considered significant.

It is crucial that the author clarify in the Discussion whether this difference in inclusion criteria affects the direct comparison of prevalence rates for both types of violence, since "being a victim" is defined much more broadly in cyberbullying than in traditional bullying in this study.

Response: We agree with the reviewer that this difference in inclusion criteria affects the direct comparison of prevalence rates for both types of violence. We have clarified in the Discussion that this difference in inclusion criteria affects the direct comparison of prevalence rates for both types of violence. We have also pointed out that direct comparisons of prevalence rates between cyberbullying and traditional bullying were not the primary aim of the present study. Instead, the focus was on examining changes in prevalence over time within the same form of perpetration or victimization.

The author uses the Kruskal-Wallis test to compare the three waves (2014, 2018, 2022) and the Jonckheere-Terpstra test for ordered trends.

Although these tests are robust for nonnormal distributions, they are limited in detecting complex interactions (e.g., Year ≥ Gender ≥ Age) that are central to the study hypothesis.

Response: The Kruskal–Wallis test was used to examine differences among the 2013/14, 2017/18, and 2021/22 HBSC surveys. The Jonckheere–Terpstra test was applied to assess the presence of a statistically significant trend in the data (i.e., an ordered pattern across survey waves). Neither the Kruskal–Wallis test nor the Jonckheere–Terpstra test was intended to detect interaction effects. To our knowledge, the Kruskal–Wallis test is the nonparametric equivalent of a one-way ANOVA and therefore involves only a single independent variable; testing interaction effects would require at least two independent variables. Although the examination of complex interactions may be of interest when supported by a clear theoretical rationale, such analyses were beyond the scope of the present study and were not among its objectives.

The manuscript analyzes each subgroup separately, inflating the Type I error (although it uses Holm correction).

Response: Thank you for raising this point. We have addressed this issue in the Discussion section.

Round 2

Reviewer 1 Report

Comments and Suggestions for Authors

The authors have responded very thoughtfully and effectively to all of my comments. The revised manuscript is much strengthened and will be a valuable addition to the literature.

Reviewer 2 Report

Comments and Suggestions for Authors

no comment